# Interpretable Sparse System Identification: Beyond Recent Deep Learning Techniques on Time-Series Prediction

**Xiaoyi Liu[1], Duxin Chen[1]\*, Wenjia Wei[2], Xia Zhu[2] & Wenwu Yu[1]**
[1] School of Mathematics, Southeast University, Nanjing 210096, China
[2] Network Laboratory, 2012 Lab, Huawei Technologies Co. Ltd, Shenzhen 518129, China
`{xiaoyiliu,chendx,wwyu}@seu.edu.cn`
`{weiwenjia,zhuxia1}@huawei.com`

## Abstract

With the continuous advancement of neural network methodologies, time series prediction has attracted substantial interest over the past decades. Nonetheless, the interpretability of neural networks is insufficient and the utilization of deep learning techniques for prediction necessitates significant computational expenditures, rendering its application arduous in numerous scenarios. In order to tackle this challenge, an interpretable sparse system identification method which does not require a time-consuming training through back-propagation is proposed in this study. This method integrates advantages from both knowledge-based and data-driven approaches, and constructs dictionary functions by leveraging Fourier basis and taking into account both the long-term trends and the short-term fluctuations behind data. By using the $l_1$ norm for sparse optimization, prediction results can be gained with an explicit sparse expression function and an extremely high accuracy. The performance evaluation of the proposed method is conducted on comprehensive benchmark datasets, including ETT, Exchange, and ILI. Results reveal that our proposed method attains a significant overall improvement of more than 20% in accordance with the most recent state-of-the-art deep learning methodologies. Additionally, our method demonstrates the efficient training capability on only CPUs. Therefore, this study may shed some light onto the realm of time series reconstruction and prediction.

## 1 Introduction

Long-term prediction with less input data plays a crucial role in the contemporary era of big data. Numerous domains, including electricity, exchange-rate, and disease (Wu et al. (2021); Zhou et al. (2022); Zhang & Yan (2022); Zeng et al. (2023); Bi et al. (2023)), necessitate accurate predictions over long time horizons. Recently, the advancement of deep learning has led to the prominence of neural network methods, particularly Transformer-based approaches (Vaswani et al. (2017)) such as Crossformer (Zhang & Yan (2022)) and FEDformer (Zhou et al. (2022)). Nevertheless, the heavy reliance on GPU resources in deep learning has turned time series prediction into a competition of computational capabilities, rendering its challenge for ordinary laboratory setups or personal computers to cope with the demands (Wu et al. (2021)). Is it truly inconceivable to achieve long-term forecasting without the utilization of neural networks? This study endeavors to explore an innovative approach rooted in system identification, which ensures compatibility with CPU-based execution while maintaining a high level of accuracy comparable to that of deep learning methods.

Traditional time series prediction methods like ARIMA (Shumway et al. (2017)), SVM (Sapankevych & Sankar (2009)) and Temporal Regularized Matrix Factorization(TRMF, Yu et al.

---

\*Duxin Chen is the corresponding author. This work is supported by the National Key R&D Program of China under Grant No. 2022ZD0120003, the Zhishan Youth Scholar Program, the National Natural Science Foundation of China under Grant Nos. 62233004, 62273090, 62073076, the Huawei Technical Cooperation Project under Grant No. TC20220329025, and the Jiangsu Provincial Scientific Research Center of Applied Mathematics under Grant No. BK20233002.

(2016)) are some commonly used classical statistical algorithms. However, these methods are typically limited to make short-term predictions and encounter challenges when applied to long-term prediction tasks due to the accumulation of errors over time. The introduction of deep learning methods can be traced back to the emergence of RNNs models, which leverage their hidden states to summarize and retain past information. Prominent examples of RNN-based architectures include LSTM (Hochreiter & Schmidhuber (1997)), GRU (Chung et al. (2014)), and LSTNet (Lai et al. (2018)). Additionally, there are deep learning approaches that build upon classical algorithms, such as DeepGLO (Sen et al. (2019)) and Temporal Latent Auto-Encoder (TLAE, Nguyen & Quanz (2021)) , which integrate deep learning techniques into the TRMF framework for enhanced prediction performance. Furthermore, there exist many research endeavors that build upon the concept of Temporal Convolutional Networks (TCN, Aksan & Hilliges (2019); Yan et al. (2020); Lea et al. (2017)). These approaches employ a stacked arrangement of one-dimensional convolutional layers and residual connections to effectively capture both the local and global characteristics inherent in time series data. Despite demonstrating advancements over traditional methods in short-term prediction, these algorithms still face limitations when it comes to accurate long-term prediction tasks.

In recent years, the advent of the Transformer (Vaswani et al. (2017)) has given rise to a series of Transformer-based models that tackle the challenges associated with long-term prediction, continually pushing the boundaries of state-of-the-art performance in this domain. These examples include Crossformer (Zhang & Yan (2022)), FEDformer (Zhou et al. (2022)), Autoformer (Wu et al. (2021)), Informer (Zhou et al. (2021)), Pyraformer (Liu et al. (2021)), LogTrans (Li et al. (2019)), and so on. Despite efforts to optimize algorithmic complexity, it remains evident that current long-term prediction methods necessitate significant computational resources. Moreover, as the prediction horizon increases, the training time becomes intolerable for small-scale laboratories. Accordingly, our objective is to embrace a lightweight approach wherein the computational requirements are reduced while upholding a high prediction accuracy. Successful instances of this approach are exemplified by DLinear (Zeng et al. (2023)) and TiDE (Das et al. (2023)), which deviate from the conventional Transformer framework and achieve remarkable performance solely by employing simple linear models, thereby mitigating training time. Building upon this foundation, we attempt to abandon deep learning frameworks and explore a machine learning-based approach that enables long-term time series prediction on CPUs. Our goal is to develop a method that guarantees precision while remaining insensitive to the increase in prediction horizon, meaning that the training time of the model does not significantly increase as the prediction horizon grows.

Thus, we here propose the Global-Local Identification and Prediction (GLIP) model, which combines the framework of identification models (Brunton et al. (2016); Yuan et al. (2019); Gao & Yan (2022)) with insights from deep learning training methods. Specifically, The prediction model utilizes system identification methods and involves two main stages of identification and prediction. In the first stage of identification, three types of basis functions are constructed using Discrete Fourier Transform (Lu et al. (2021)) for prediction. These include global basis obtained from the training set, stored basis, and local basis from the test set. Note that prediction with explicit model instead of an entire black-box neural network approach, can also be referred to N-BEATS (Oreshkin et al. (2019)). In the second stage of identification, we further explore the relationships between variables through identification to improve prediction performance. In the entire procedure, it is apparent that GLIP abstains from employing any back-propagation mechanisms, opting instead for exclusive reliance on elementary machine learning techniques. Occam's Razor (Blumer et al. (1987)) told us Entities should not be multiplied without necessity. For the same prediction task, our model is obviously more concise and efficient. The primary contributions can be summarized as follows:

**1)**. A novel identification model for long-term prediction is introduced, which operates efficiently on CPUs instead of relying on GPUs. Additionally, the training time of the GLIP model remains largely unchanged as the prediction horizon increases. This significantly expands the applicability of the model, enabling long-term prediction even on personal computers. By offering a fresh direction amidst the prevalence of Transformer-based models in the realm of long-term prediction, this research opens new avenues for exploration.

**2)**. The utilization of Fourier transform-based basis functions in global identification and local prediction allows for precise capture of both the knowledge of long-term trends and short-term fluctuations in time series data. This approach integrating both knowledge and data driven ways effectively addresses the limitation of neural networks in a global perspective for prediction tasks.

**3)**. In the context of local basis function prediction, the incorporation of Fourier transforms as part of the basis for each local segment data enables accurate extraction of potential future trends even when the input-output ratio is very small. This capability surpasses the limitations of traditional sparse identification methods.

**4)**. Remarkable prediction performance was achieved on the four benchmark datasets, surpassing the current state-of-the-art results. The proposed model exhibited a notable 23.64% improvement in Mean Squared Error (MSE) compared to neural network approaches. This breakthrough highlights the efficacy of non-neural network models in long-term prediction tasks.

## 2 NOTATIONS

When performing time-series prediction, the predicted multivariate time series can be represented as $\mathbf{X}$. Here, $\mathbf{X}$ is a two-dimensional vector, where the number of rows corresponds to the count of variables, and the number of columns represents the length of the time series. The $i$-th row of $\mathbf{X}$ is denoted as $\mathbf{X}[i,:]$, while the $i$-th column of $\mathbf{X}$ is denoted as $\mathbf{X}[:,i]$. Furthermore, the subset of rows (columns) from the $i$-th to $j$-th positions in $\mathbf{X}$ can be denoted as $\mathbf{X}[i:j,:]$ ($\mathbf{X}[:,i:j]$). Matrices with the same number of rows or columns can be concatenated either horizontally or vertically. The horizontal concatenation is denoted as $[\mathbf{X}, \mathbf{Y}]$, while the vertical concatenation is denoted as $[\mathbf{X}; \mathbf{Y}]$. For instance, time series X comprising $m$ variables can be represented as $[x_1; x_2; \cdots; x_m]$, where $x_i$ denotes a specific time series. The linspace function is used to create a vector with equally spaced elements. $\text{linspace}(x_1, x_2, n)$ generates a row vector containing $n$ elements that are evenly spaced between $x_1$ and $x_2$. The symbol $\circ$ represents the Hadamard product, which denotes the element-wise multiplication of corresponding elements in vectors or matrices. A scalar can be added to or subtracted from a vector. When a scalar is added to or subtracted from a vector, the scalar will undergo the corresponding operation with each element of the vector. In addition, if we want to explore higher-order relationships among variables in a multivariate matrix $\mathbf{X}$, such as a quadratic relationship, we denote it as $\mathbf{X}^2$, which signifies:

$$[\mathbf{X}[1,:]^2; \mathbf{X}[1,:]\mathbf{X}[2,:]; \cdots \mathbf{X}[1,:]\mathbf{X}[m,:]; \mathbf{X}[2,:]^2; \mathbf{X}[2,:]\mathbf{X}[3,:]; \cdots \mathbf{X}[m,:]^2].$$

In addition, this paper also distinguishes between global prediction and local prediction, the significance of which can be further explored in Appendix. A.

## 3 METHODOLOGY

In this section, a thorough exposition of the proposed Global-Local Identification and Prediction (GLIP) model will be presented. Analogous to deep learning methodologies, the time series will be partitioned into distinct sets, which are training set, validation set, and testing set, with a partitioning ratio of 7:1:2. In the stage of the entire prediction framework, we first construct global basis and storage basis in the training set for global prediction. Secondly, we evaluate the performance of global prediction in the validation set to determine its suitability for local rolling prediction. Finally, in the test set, the amalgamation of all preceding information will be leveraged to execute local rolling prediction. The detailed procedures of global prediction and local rolling prediction are illustrated in Figure 1.

### 3.1 GLOBAL IDENTIFICATION

The propose of global identification is to investigate the macroscopic variations exhibited in the given time series. We will employ Fourier transformation and sparse identification techniques on the training set. This approach facilitates the extraction and design of the global basis and storage basis, which serve as preparatory components for subsequent local rolling identification. Furthermore, the utilization of the global basis enables a preliminary estimation of global trend, thereby enabling long-term predictions surpassing 1000 data points.

#### 3.1.1 GLOBAL BASIS AND GLOBAL PREDICTION

Global identification involves initially applying Fourier transformation to seek potential frequencies or periods behind the training data. Subsequently, a global basis is constructed by utilizing these

inferred periods exhibited in the frequency domain to form a basis of trigonometric functions. Sparse identification is then performed using this global basis, followed by global prediction. Specifically, based on the Discrete Fourier Transformation (DFT), denoted as: $\mathbf{A}, \mathbf{f} = \mathrm{DFT}(\mathbf{X})$, where $\mathbf{A}, \mathbf{f}$ are amplitude and frequency of DFT, respectively. For further global prediction, frequency with high amplitude are selected (see Appendix. C). By utilizing the frequency-period formula $f = 1/T$, these frequencies are converted into corresponding periods, unveiling the underlying potential periodicity $\mathbf{T}^* = [T_1, T_2, ...]$. These potential periods are employed to construct the global basis function

$$\mathbf{\Theta}_g = [\sin(\mathbf{C}_1 \mathbf{t}), \cos(\mathbf{C}_1 \mathbf{t}), \mathbf{1}], \tag{1}$$

where $\mathbf{C}_1 = 2\pi/\mathbf{T}^*$ and $\mathbf{t}$ is a column vector represent the time of train length. The following sparse identification process can be performed using the global basis.

$$\underset{\mathbf{\Xi}_g}{\arg\min} \int_0^T \left( \|\mathbf{X} - \mathbf{\Theta}_g \mathbf{\Xi}_g\|^2 \right) \mathrm{d}\mathbf{t} + \lambda \|\mathbf{\Xi}_g\|. \tag{2}$$

There exist several well-established algorithms for solving the aforementioned optimization problem (equation 2), such as OMP (Schnass (2018)), LASSO (Ranstam & Cook (2018)), SBM (Jacobs et al. (2018)), and others. In this context, we employ the $l_1$ norm for sparsity regularization and utilize the typical coordinate descent method similar to LASSO for optimization. After obtaining $\mathbf{\Theta}_g$, one can set $t = L_{\mathrm{train}} + 1, L_{\mathrm{train}} + 2, ...(L_{\mathrm{train}}$ is the length of training set) to predict the data in the validation and test sets, recording the predicted result as $\mathbf{X}_{gp}$. Furthermore, to capture the global inter-dependency among variables, we can concatenate the results obtained from the univariate predictions with the training set, denoting as $\mathbf{X}_g = [\mathbf{X}[:, : \mathrm{train}], \mathbf{X}_{gp}]$. Thus constructing a new basis function library $\mathbf{\Theta}_g^*$ for further network relationship identification, i.e.,

$$\mathbf{\Theta}_g^* = [\mathbf{X_g}, \mathbf{X}_g^2, \mathbf{X}_g^3, ..., \sin(\mathbf{X}_g), \cos(\mathbf{X}_g), ...]. \tag{3}$$

By substituting $\mathbf{\Theta}_g^*$ into equation 2 and optimizing it, we can obtain the updated variable predictions $\mathbf{X}_{gp}^*$ with network coupling relationships. This represents the outcome of global identification and prediction. Appendix. B describes the introduction to system identification.

The schematic diagram of the process for establishing the global basis can be seen in the upper part of the "Global Prediction" section in Figure 1. The global identification and prediction presents two notable advantages. Firstly, sparse identification acts as a robust mechanism to mitigate the risk of overfitting, facilitating the discernment of authentic underlying patterns or latent cycles. Notably, the direct utilization of the extracted high-frequency feature by DFT for reconstruction restricts the identification efficacy to the training set, with limited the generalizability to the validation and prediction sets. Secondly, this module, by exclusively employing $t$ for prediction during the initial identification phase, circumvents the common issue of error accumulation in long-term prediction encountered by traditional identification methods.

### 3.1.2 STORAGE BASIS

The storage basis serves as a preparatory foundation for local rolling prediction. Given that this model does not employ backpropagation for parameter updates, when conducting local rolling prediction on the test set, it is challenging to fully leverage the data feature from the training set, except for the solely obtained results of global identification and prediction. Considering our primary focus on prediction from an identification perspective, it is crucial to construct a set of basis, known as the storage basis, that is simultaneously relevant to both the training set and local rolling prediction.

If the length of a batch is denoted as $L_{\mathrm{batch}}$ (i.e., the length of input and output sequences), we continuously slide a window of length $L_{\mathrm{batch}}$ over the test set with an interval of $L_{\mathrm{interval}}$. During each sliding window, we perform the DFT on each variable of the small window's time series and record the high-frequency components in the frequency domain. We utilize a set to store these high-frequency components to ensure there is no duplication. Ultimately, this process yields a set of periods $\mathbf{T}_s^*$. Leveraging these periods, similar to the global basis, we can construct the following storage basis:

$$\mathbf{\Theta}_s = [\sin(\mathbf{C}_2 \mathbf{t}), \cos(\mathbf{C}_2 \mathbf{t})], \tag{4}$$

where, $\mathbf{C}_2 = 2\pi/\mathbf{T}_s^*$. The illustration of the storage basis is located below the "Global Prediction" section in Figure 1. The storage basis functions will have a significant impact on local rolling prediction, addressing the challenge faced by traditional methods in effectively using information from the training set during local rolling prediction.

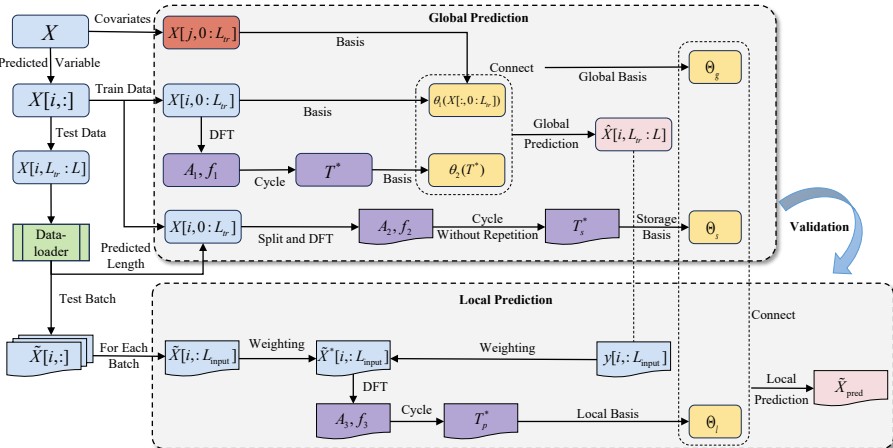

Figure 1: The GLIP model architecture

## 3.2 VALIDATION IDENTIFICATION

The objective of validation identification is to evaluate the efficacy of conducting local rolling prediction on the validation set, utilizing two global identification mentioned in the previous sections as a reference. This assessment serves as a preparatory step for subsequent local rolling prediction.

For the sake of simplicity and clarity, we denote $\tilde{\mathbf{X}}$ as a generic representation of a batch, where $\tilde{\mathbf{X}}$ refers to $\mathbf{X}[:, i : i + L_{\text{batch}}]$. Meanwhile, we assume that the output derived from the local rolling prediction model is denoted as $\tilde{\mathbf{X}}_{\text{pred}}$. Its significance is similar to that of $\tilde{\mathbf{X}}$, representing a generic representation of a batch.

### 3.2.1 GLOBAL IDENTIFICATION VALIDATION

Validating global identification is to compare whether utilizing local features for prediction yields superior performance compared to global prediction. Assume that the predicted values of global prediction for a specific batch are denoted as $\tilde{\mathbf{X}}_{\text{global}}$. To evaluate the quality of global prediction, it is necessary to observe whether there is a significant improvement in prediction accuracy for each variable $i$ when utilizing local features. i.e.,

$$\frac{1}{N_b} \sum_{N_b} \left| \frac{\tilde{X}_{\text{global}}[i, L_{\text{input}} :] - \tilde{X}[i, L_{\text{input}} :]}{\tilde{X}_{\text{pred}} - \tilde{X}[i, L_{\text{input}} :]} \right| \leq k_1, \tag{5}$$

where $N_b$ is the number of validation batches. $k_1$ is a hyperparameter, and it can be adjusted based on the desired level of improvement. In the subsequent local rolling prediction, we will determine whether it is necessary to further incorporate local basis prediction by observing whether the variables satisfy equation 5.

### 3.2.2 VARIABLES RELATIONSHIP VALIDATION

The validation of variable relationships aims to verify whether, after conducting local rolling predictions, the coupling relationships between different variables can further enhance the prediction effectiveness. To accomplish this, assuming that the output of coupling relationships after local rolling prediction is $\tilde{\mathbf{X}}^*$. Meanwhile, we introduce a new hyperparameter, $k_2$. By substituting $\tilde{\mathbf{X}}^*$ for $\tilde{\mathbf{X}}_{\text{global}}$, replacing $k_1$ with $k_2$ and swapping the numerator and denominator in the summation symbol in equation 5, we can use the modified equation 5 to determine whether further coupling of variables is needed in the test set to improve prediction performance.

### 3.3 LOCAL IDENTIFICATION

This section will primarily focus on local identification and utilize the results obtained from global identification and validation identification to enhance the shortcomings of traditional methods in prediction. We aim to achieve local rolling prediction through local identification.

#### 3.3.1 LOCAL IDENTIFICATION CURVE

In the context of local rolling prediction, both conventional approaches and prevalent neural network methods commonly employ the raw values of the input for prediction. Nonetheless, it is important to acknowledge that the input values may exhibit outliers or abrupt changes, thereby exerting a substantial influence on the prediction accuracy. Consequently, prior to the local prediction, this section necessitates the implementation of a straightforward preprocessing procedure for the input data, which will draw upon the insights gained from global identification and prediction.

We aim to construct a new input curve that is a weighted sum of both the global prediction and the real time series. This approach allows us to capture both the global trend and local fluctuations in the data. Assuming that the input data is denoted as $x$, and the corresponding results obtained from global identification are denoted as $y$. Now, we construct a new input $x^*$ as follows:

$$x^* = w \circ x + (1-w) \circ y, \tag{6}$$

where $w = \left(\text{linspace}(\alpha^{1/\gamma}, \beta^{1/\gamma}, L_{\text{batch}})\right)^{\gamma}$. In general, $0 \leq \alpha < \beta \leq 1$. Typically, $\alpha$ and $\beta$ are determined based on the quality of global prediction. If the global prediction is good, smaller values are assigned to $\alpha$ and $\beta$, indicating a higher weight for global prediction and resulting in a prediction that leans more towards the global trend. Conversely, larger values are assigned to $\alpha$ and $\beta$, allowing more emphasis on local information. $\gamma$ reveals the pace of weight changes and is usually set to 1. By adjusting the parameters to appropriate values, we can construct the local identification curve. In ablation experiments, we have demonstrated that utilizing $x^*$ for prediction yields superior results compared to directly using $x$.

#### 3.3.2 LOCAL PREDICTION

The local prediction consists of three steps. The first step is to determine if local identification is necessary. If a variable demonstrates good performance in the validation identification, satisfying equation 5, we do not need to proceed with further prediction and can directly use the results of global prediction for local prediction. Otherwise, we proceed to the second step.

The second step involves selecting the local basis. If certain variables fail to meet the criterion specified by equation 5, it suggests that the performance of global identification is inadequate and needs further enhancements. Leveraging the locally derived identification curve obtained from equation 6, we adopt a methodology akin to global identification and conduct DFT. Subsequently, we identify the high-frequency components and transform them into potential periods denoted as $T_p^*$.

$$\mathbf{\Theta}_l = [\sin(\mathbf{C_3}\mathbf{t}), \cos(\mathbf{C_3}\mathbf{t})], \tag{7}$$

where, $\mathbf{C_3} = 2\pi/\mathbf{T}_p^*$. The "Local Prediction" section in Figure.1 illustrates the process of selecting the local basis. However, relying solely on this basis is insufficient for local rolling prediction. This is because the input for local rolling prediction is limited in terms of available data, and the absence of global information can potentially lead to significant prediction biases.

Therefore, the third step involves integrating the basis and performing local prediction. We combine the previously constructed basis with the local basis functions to obtain a comprehensive representation of global information. This incorporates three types of basis: the global basis obtained from global identification (equation 1), the storage basis constructed from additional computations on the test set (equation 4), and the local basis (equation 7) derived from the local identification curve. By merging these three types of basis, we obtain the final basis functions for local identification and prediction, i.e.,

$$\mathbf{\Theta}_{\text{pred}} = [\mathbf{\Theta}_g, \mathbf{\Theta}_s, \mathbf{\Theta}_l]. \tag{8}$$

By utilizing this integrating basis to replace $\Xi$ in equation 2 and performing sparse identification, we can obtain $\Xi_{\text{pred}}$. Letting $t = L_{\text{input}}+1, L_{\text{input}}+2, ...$, enables us to make local rolling predictions and obtain $\tilde{\mathbf{X}}_{\text{pred}}$. It is worth mentioning that $\tilde{\mathbf{X}}_{\text{pred}}$ is obtained in a similar manner during the validation identification phase.

| Models | | IMP. | GLIP | | DLinear | | FEDformer | | Autoformer | | Informer | | Reformer | | Pyraformer | | LogTrans | |
|---|---|---|---|---|---|---|---|---|---|---|---|---|---|---|---|---|---|---|
| Metric | | MSE | MSE | MAE | MSE | MAE | MSE | MAE | MSE | MAE | MSE | MAE | MSE | MAE | MSE | MAE | MSE | MAE |
| ETTh | 96 | 24.91% | **0.217** | **0.332** | 0.289 | 0.353 | 0.346 | 0.388 | 0.358 | 0.397 | 3.755 | 1.525 | 0.845 | 0.693 | 0.645 | 0.597 | 2.116 | 1.197 |
| | 192 | 34.46% | **0.251** | **0.364** | 0.383 | 0.418 | 0.429 | 0.439 | 0.456 | 0.452 | 5.602 | 1.931 | 0.958 | 0.741 | 0.788 | 0.683 | 4.315 | 1.635 |
| | 336 | 38.39% | **0.276** | **0.383** | 0.448 | 0.465 | 0.496 | 0.487 | 0.482 | 0.486 | 4.721 | 1.835 | 1.044 | 0.787 | 0.907 | 0.747 | 1.124 | 1.604 |
| | 720 | 32.18% | **0.314** | **0.408** | 0.605 | 0.551 | 0.466 | 0.474 | 0.515 | 0.511 | 3.647 | 1.625 | 1.458 | 0.987 | 0.963 | 0.783 | 3.188 | 1.540 |
| ETTm | 96 | 11.38% | **0.148** | **0.260** | 0.167 | **0.260** | 0.203 | 0.287 | 0.255 | 0.339 | 0.365 | 0.453 | 0.658 | 0.619 | 0.435 | 0.507 | 0.768 | 0.642 |
| | 192 | 17.86% | **0.184** | **0.302** | 0.224 | 0.303 | 0.269 | 0.328 | 0.281 | 0.340 | 0.533 | 0.563 | 1.078 | 0.827 | 0.730 | 0.673 | 0.989 | 0.757 |
| | 336 | 21.35% | **0.221** | **0.331** | 0.281 | 0.342 | 0.325 | 0.366 | 0.339 | 0.372 | 1.363 | 0.887 | 1.549 | 0.972 | 1.201 | 0.845 | 1.334 | 0.872 |
| | 720 | 34.00% | **0.262** | **0.360** | 0.397 | 0.421 | 0.421 | 0.415 | 0.433 | 0.432 | 3.379 | 1.338 | 2.631 | 1.242 | 3.625 | 1.451 | 3.048 | 1.328 |
| Exchange | 96 | 11.11% | **0.072** | **0.202** | 0.081 | 0.203 | 0.148 | 0.278 | 0.197 | 0.323 | 0.847 | 0.752 | 1.065 | 0.829 | 0.376 | 1.105 | 0.968 | 0.812 |
| | 192 | 20.38% | **0.125** | **0.275** | 0.157 | 0.293 | 0.271 | 0.380 | 0.300 | 0.369 | 1.204 | 0.895 | 1.188 | 0.906 | 1.748 | 1.151 | 1.040 | 0.851 |
| | 336 | 31.48% | **0.209** | **0.348** | 0.305 | 0.414 | 0.460 | 0.500 | 0.509 | 0.524 | 1.672 | 1.036 | 1.357 | 0.976 | 1.874 | 1.172 | 1.659 | 1.081 |
| | 720 | 31.10% | **0.443** | **0.516** | 0.643 | 0.601 | 1.195 | 0.841 | 1.447 | 0.941 | 2.478 | 1.310 | 1.510 | 1.016 | 1.943 | 1.206 | 1.941 | 1.127 |
| ILI | 24 | 18.65% | **1.802** | **0.908** | 2.215 | 1.081 | 3.228 | 1.260 | 3.483 | 1.287 | 5.764 | 1.677 | 4.366 | 1.382 | 1.420 | 2.012 | 4.480 | 1.444 |
| | 36 | 9.52% | **1.766** | **0.943** | 1.963 | 0.963 | 2.679 | 1.080 | 3.103 | 1.148 | 4.755 | 1.467 | 4.446 | 1.389 | 7.394 | 2.031 | 4.799 | 1.467 |
| | 48 | 18.73% | **1.731** | **0.952** | 2.130 | 1.024 | 2.622 | 1.078 | 2.669 | 1.085 | 4.763 | 1.469 | 4.572 | 1.436 | 7.551 | 2.057 | 4.800 | 1.468 |
| | 60 | 22.80% | **1.828** | **0.992** | 2.368 | 1.096 | 2.857 | 1.157 | 2.770 | 1.125 | 5.264 | 1.564 | 4.743 | 1.487 | 7.662 | 2.100 | 5.278 | 1.560 |

Table 1: Local rolling prediction errors in terms of MSE and MAE. The results from other models are sourced from (Zeng et al. (2023); Wu et al. (2021); Kitaev et al. (2020)). For ILI, the input length for local rolling prediction is 24, with output lengths 24, 36, 48, 60. For other datasets, the input length is 96, with output lengths 96, 192, 336, 720. The best results are indicated by **bold numbers** in the table, while the second-best results are marked with horizontal lines.

### 3.3.3 LOCAL VARIABLES RELATIONSHIP

Lastly, we consider the relationships among different variables in local prediction. During the validation identification phase, we assess which variables are suitable for representation using other variables as basis. We extract these variables and construct basis similar to equation 2. Similar to the global identification process, we perform a second identification using these basis. By utilizing the identification results, we can derive further prediction results for each variable. This predicted outcome, denoted as $\tilde{\mathbf{X}}_{\text{pred}}^{*}$, represents the final prediction result for local rolling prediction. In Appendix. E, we provide pseudo-code for the entire algorithmic process.

## 4 EXPERIMENTS

We will show the results of global prediction and local rolling prediction. Furthermore, we will highlight the advantages of GLIP in terms of computational efficiency and long-term prediction compared to neural networks. The experimental settings can be found in Appendix. D.

### 4.1 PERFORMANCE COMPARISON

### 4.1.1 LOCAL ROLLING PREDICTION

As shown in Table. 1, GLIP achieved excellent results in local rolling prediction across the four benchmark datasets. Specifically, it achieved a 32.49% MSE improvement on the ETTh dataset, a 21.14% MSE improvement on the ETTm dataset, a 23.52% MSE improvement on the Exchange dataset, and a 17.42% MSE improvement on the ILI dataset. Overall, compared to neural network methods, GLIP achieved a total improvement of 23.64%. We observed that the improvement tends to be greater when the O/I is larger. For instance, when the O/I is 7.5, the MSE improvement is consistently above 30%. This indicates that GLIP has the ability to predict long-term data based on short-term data in local rolling prediction, addressing the challenge of long-term prediction that traditional methods cannot handle. Figure 2 (d)-(f) showcase the visualization results of the three datasets when performing local rolling prediction with the longest output.

It is noteworthy that in the context of local rolling prediction, GLIP attains remarkable performance by exclusively employing a composite of Fourier basis, subsequently coupled with sparse identification. While sharing some conceptual resemblance with FEDformer-f (Zhou et al. (2022)), GLIP distinctively circumvents the reliance on a neural network architecture and instead directly conducts

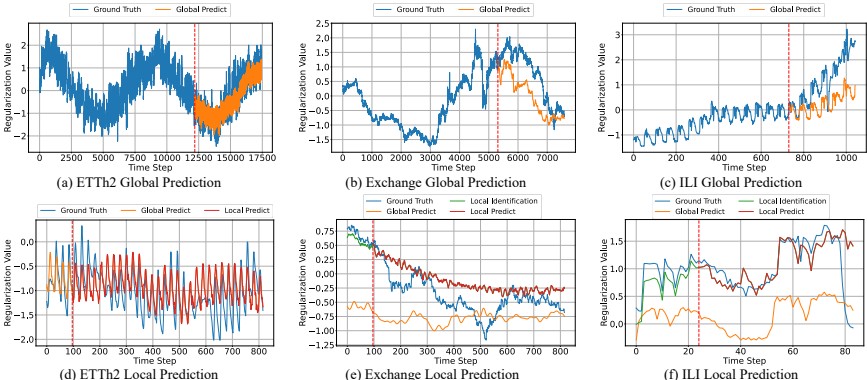

Figure 2: The visualization results depict the prediction of a specific variable in the ETT, Exchange, and ILI datasets. Figures (a), (b), and (c) represent global predictions, while Figures (d), (e), and (f) illustrate a slice of local rolling predictions. The labels in the figure hold the following implications: "Ground Truth" signifies the veritable time series data; "Global Prediction" denotes the visual representation of global prediction, aligning with the output outcomes from the global identification module; "Local Identification" represents the result of local identification after the weighted combination of global predictions and local input data; "Local Prediction" encompasses the outcomes of local rolling prediction; and the red vertical line demarcates the division between the training set and test set.

the prediction, thereby yielding superior outcomes. This observation underscores the viability of non-neural network methodologies for time series prediction in certain scenarios.

### 4.1.2 GLOBAL PREDICTION

Global prediction is a distinctive advantage of GLIP, which sets it apart from neural network-based methods that necessitate an ample number of training samples and are therefore incapable of accomplishing global prediction. This implies that neural networks may not consistently capture the genuine trends of time series from a macroscopic perspective. In certain scenarios, GLIP can provide a reference for global prediction, as depicted in Figure 2 (a)-(c). However, it is important to note that global predictions may not always be accurate and further local predictions may be required.

An example of accurate global prediction is the oil temperature variable (OT) in ETTh and ETTm, as illustrated in Figure 2 (a). Through identification, we have observed that global identification and prediction yield better results compared to local rolling prediction. Therefore, in local rolling prediction, we directly utilize the results of global prediction as depicted in Figure 2 (d).

### 4.1.3 ENVIRONMENT AND EFFICIENCY

Considering that GLIP only uses a few machine learning techniques for implementation, it does not require a GPU environment for experiment. Each of the aforementioned experiments can be completed in a matter of seconds to minutes on a personal computer running on a CPU environment. This greatly reduces our computational requirements. According to the principle of Occam's razor, GLIP undoubtedly stands as a superior prediction model.

Furthermore, considering the structure of GLIP, its efficiency in learning the local rolling prediction model is essentially independent of the prediction length. Once the model structure is determined, it can directly perform predictions without being affected by the increase in prediction length. Specifically, the time required for predictions with a length of 720 is comparable to that of predictions with a length of 96 (see Supplementary Material).

### 4.1.4 INTERPRETABILITY AND PARAMETER ADJUSTMENT

The interpretability of GLIP lies in the following three aspects. Firstly, GLIP employs DFT to explicitly identify potential periods within the time series for both global and local identification,

incorporating them into the basis functions. This can be utilized to interpret the main components extracted from the frequency domain of the time series. Secondly, in comparison with neural networks, GLIP has an explicit mathematical function expression for time series prediction tasks. The function expression provides insights into the predicted trends with an explicit white-box form. Thirdly, the hyperparameters, namely $k_1, k_2, \alpha, \beta, \lambda$, and parameters $\Xi$ have practical significance, e.g., $\Xi$ indicates the cycles and magnitudes between variables, which can be used directly to interpret the model. Comparatively, parameters such as the number of hidden layers and neurons in neural networks or the number of multi-heads in Transformer may not have straightforward interpretability. Furthermore, owing to the interpretability of the parameters and the expeditious computational efficiency of the model, the process of parameter adjustment is markedly simpler in contrast to neural networks (see Appendix. D).

## 4.2 ABLATION EXPERIMENT

While GLIP consists of several components integrated together for prediction, it is crucial to emphasize that each component has been meticulously constructed and holds irreplaceable significance. Removing or altering any of these components may potentially render the identification process ineffective. To demonstrate the significance of these components, we excluded different components and performed ablation experiment. The validation results on the Exchange dataset are presented in Table 2.

| $L_{\text{output}}$ | Metric | GLIP | Case 1 | Case 2 | Case 3 |
|---|---|---|---|---|---|
| 96 | MSE | **0.072** | 1.912 | 0.087 | 0.091 |
| | MAE | **0.202** | 1.151 | 0.212 | 0.220 |
| 192 | MSE | **0.125** | 1.186 | 0.206 | 0.232 |
| | MAE | **0.275** | 1.135 | 0.328 | 0.318 |
| 336 | MSE | **0.209** | 1.182 | 0.485 | 0.792 |
| | MAE | **0.348** | 1.122 | 0.496 | 0.453 |
| 720 | MSE | **0.443** | 1.759 | 0.602 | 2.450 |
| | MAE | **0.516** | 1.014 | 0.599 | 0.757 |

Table 2: Results for ablation experiments on the Exchange dataset. Case 1 omits sparse identification, Case 2 lacks local identification curves, and Case 3 excludes storage basis.

**Effect of Sparse Identification:** If high-frequency Fourier bases are directly used for prediction without undergoing sparse identification, it is possible that the resulting predictions on the training set will closely resemble the original dataset. However, this approach may lead to overfitting. We refer to the absence of the sparse identification component as Case 1. From Table 2, it is evident that Case 1 has no predictive capability.

**Effect of Local Identification Curve:** During the process of local rolling identification, if the input values are directly utilized for prediction, it may overlook the global trend information. Additionally, it is highly probable to encounter overfitting issues when dealing with input points that exhibit an abrupt or anomalous behavior, consequently leading to prediction errors. The absence of local identification curves is denoted as Case 2. From Table 2, it can be noted that Case 2 exhibits weaker predictive capabilities compared to GLIP.

**Effect of Storage Basis:** Storage bases are crucial components in prediction tasks as they maximize the utilization of information provided by the training set. Without storage bases, our training is confined to a local context, limiting the ability to leverage historical information from a global perspective. Case 3 is designated to represent the scenario where trend storage bases are absent. From Table 2, it can be seen that Case 3 demonstrates close proximity to GLIP in short-term predictions, but loses its predictive capability in long-term prediction.

## 5 CONCLUSION AND FUTURE WORK

In this study, we have provided a detailed account of the GLIP model to global prediction and local rolling prediction. By ingeniously constructing basis functions extracting knowledge from the frequency domain as much as possible and employing sparse identification techniques, we are able to accomplish long-term prediction tasks within a CPU environment. This breakthrough overcomes the limitations of traditional methods with a heavy computational burden and opens up a new direction for long-term prediction.

In future work, we plan to enhance both the scale of the data and the scope of the predictions. Additionally, we aim to integrate the identification methods with neural network approaches, striving to achieve superior prediction performance within shorter time frames.

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

## A  GLOBAL AND LOCAL PREDICTION OF TIME SERIES

Time series prediction can be broadly classified into global prediction and local rolling prediction. Global prediction involves predicting the entire time series by partitioning the temporal data into distinct sets, namely the training set, validation set, and testing set. Global Prediction (GP) model utilizes the training and validation sets to directly predict the remaining whole data points within the time series. However, GP models (especially neural network models) usually failed when confronted with lengthy time series due to limited availability of training samples. Conversely, local rolling prediction has emerged as a prevalent methodology in contemporary deep learning practices. This way entails segmenting the training, validation, and testing sets into multiple batches, typically representing localized segments within the time series, with each batch comprising input (training) and output (prediction) segments. The output-input ratio is denoted as O/I. In the context of local prediction, an O/I ratio larger than 1 is generally deemed indicative of long-term prediction. Presently, several models (Wu et al. (2021); Liu et al. (2021); Zeng et al. (2023)) employed for long-term prediction within local rolling settings have achieved notable O/I ratios of 7.5. However, despite such advancements, the achievable prediction length often remains limited to less than 1000 due to inherent constraints associated with local predictions. In contrast, global prediction scenarios frequently permit prediction lengths surpassing 1000 or even larger for extended time series. Currently, almost no methods exist to effectively address the challenges presented by these scenarios.

## B  INTRODUCTION TO SPARSE SYSTEM IDENTIFICATION

Sparse system identification (Wilms et al. (2023); Fasel et al. (2022); Brunton et al. (2016)) is a data-driven methodology employed for the inference of dynamic equations or algebraic equations governing a system based on observed data. The core idea behind sparse identification is to first establish a basis function library, typically polynomial series, trigonometric sequence etc. By minimizing the sparsity penalty term within the equation, the objective is to identify equations from the basis function library that possess the fewest non-zero terms. This process facilitates the elimination of extraneous terms, thereby engendering a more succinct and comprehensible dynamic model. The fundamental formulation of sparse identification is represented as

$$\dot{\mathbf{X}} = \mathbf{\Theta}(\mathbf{t}, \mathbf{X})\mathbf{\Xi} + \mathcal{N}, \tag{9}$$

where, $\mathbf{\Theta}$ represents the basis function library, $\mathbf{t}$ is the prediction time, $\dot{\mathbf{X}}$ represents the 1, 2, ..., $l$-step data of all systems before the prediction data in the $l$-order dynamic equation, and $\mathcal{N} \sim N\left(0, \sigma^2 \mathbf{I}\right)$. In some instances, sparse identification successfully discerns the latent system dynamics concealed within the data, yielding outcomes of remarkable precision (Gao & Yan (2022); Yuan et al. (2019)). Nonetheless, direct utilization of this method with no strategies on designing the basis functions to capture the knowledge behind data may engender cumulative errors, thereby rendering it unsuitable for long-term prediction. Consequently, it becomes imperative to develop innovative identification models capable of addressing the challenges associated with long-term identification and prediction.

## C  INTRODUCTION TO DISCRETE FOURIER TRANSFORM (DFT)

Discrete Fourier Transformation (DFT) can be denoted as:

$$\mathbf{A}, \mathbf{f} = \text{DFT}(\mathbf{X}).$$

The amplitude $\mathbf{A}$ and frequency $\mathbf{f}$ obtained through DFT are typically utilized for analyzing the frequency domain characteristics of time series. $\mathbf{A}$ and $\mathbf{f}$ are vectors of equal length, maintaining a one-to-one correspondence. Generally speaking, the larger the amplitude, the greater the impact of the corresponding frequency on the time series.

In this instance, Our objective is to select, based on the amplitude magnitude, the frequencies most likely to construct the main components of the time series. Subsequently, these selected frequencies are transformed into basis functions for consideration, which will be further screened through sparse identification. Therefore, our candidate periodical terms $\mathbf{T}^*$ are determined by the selection process involving $\mathbf{A}$ and $\mathbf{f}$.

# D EXPERIMENTAL SETTINGS

## D.1 DATASETS

We selected four benchmark datasets, namely ETTh, ETTm, Exchange, and ILI (Zeng et al. (2023)), for validation and testing purposes. These datasets are wildely used and consist of multivariate time series data that cover various real-world domains such as temperature, exchange, and disease. One can refer to (Wu et al. (2021); Zeng et al. (2023)) for detailed descriptions of these data.

## D.2 BASELINES

Considering the limitations of traditional methods in long-term predictions, our comparative analysis primarily focuses on neural network-based methods. Specifically, we concentrate on local rolling prediction and compare it with several prominent models, including DLinear (Zeng et al. (2023)), FEDformer (Zhou et al. (2022)), Autoformer (Wu et al. (2021)), Informer (Zhou et al. (2021)), Reformer (Kitaev et al. (2020)), Pyraformer (Liu et al. (2021)), and LogTrans (Li et al. (2019)). In instances where multiple models are available for a specific method (e.g., DLinear and FEDformer), we select the model with higher accuracy for the comparative evaluation. We will provide a concise overview of these models:

**1) DLinear** (Zeng et al. (2023)) is a simple neural network prediction model that combines time series decomposition and linear layers. It is a typical network model that operates independently of the Transformer framework. It is worth noting that recent models such as Crossformer (Zhang & Yan (2022)) shares similar levels of accuracy with DLinear. To maintain brevity in the discussion, a detailed experimental comparison will not be presented in this paper. The source code is available at `https://github.com/cure-lab/LTSFLinear`.

**2) FEDformer** (Zhou et al. (2022)) is a Transformer-based model that addresses long-term forecasting tasks by utilizing Fourier and wavelet bases. FEDformer is characterized by a time complexity of $\mathcal{O}(L)$, where $L$ represents the sequence length, and its implementation code can be found at `https://github.com/MAZiqing/FEDformer`.

**3) Autoformer** (Wu et al. (2021)) is a Transformer-based model that incorporates autocorrelation mechanisms to capture the inherent autocorrelation structure within a sequence. By introducing autocorrelation mechanisms, the model can effectively leverage the periodic information within the time series, thereby enhancing prediction accuracy. The source code is available at `https://github.com/thuml/Autoformer`.

**4) Informer** (Zhou et al. (2021)) is a Transformer-based model with ProbSparse self-attention. The time complexity of Informer is $\mathcal{O}(L \log L)$. The source code is available at `https://github.com/zhouhaoyi/Informer2020`.

**5) Pyraformer** (Liu et al. (2021)) is a Transformer-based model that introduces pyramid attention mechanisms for information propagation. This model has a time complexity of $\mathcal{O}(L)$. The source code of Pyraformer can be found at `https://github.com/alipay/Pyraformer`.

**6) Reformer** (Liu et al. (2021)) is a Transformer-based model that incorporates Locality Sensitive Hashing (LSH) attention mechanism and RevNet to reduce computational complexity. This model has a time complexity of $\mathcal{O}(L \log L)$, where L represents the sequence length. The source code of Reformer can be found at `https://github.com/google/trax/tree/master/trax/models/reformer`.

**7) LogTrans** (Li et al. (2019)) is a modification of the Transformer model that incorporates Logsparse attention and Restart attention mechanisms to reduce computational complexity. By utilizing these mechanisms, LogTrans achieves a time complexity of $\mathcal{O}(L(\log L)^2)$, where L represents the sequence length. The source code for LogTrans is avaliable at `https://github.com/mlpotter/Transformer_Time_Series`.

## D.3 EVALUATION METRICS

To ensure consistency with the aforementioned approach, we still employ the Mean Absolute Error (MAE) and Mean Square Error (MSE) evaluation metrics to determine the predictive accuracy of the time series. Furthermore, we ensure complete alignment with the aforementioned approaches

regarding the proportions of the training, validation, and test sets, as well as the input and output lengths. In Table 1, we present the improvement in MSE. Here, we briefly provide the average MAE improvement for the four datasets: ETTh: 15.6%, ETTm: 4.51%, Exchange: 9.18%, ILI: 8.65%.

### D.4 Hyperparameters adjustment

GLIP provides default tuning methods and manual tuning methods regarding hyperparameter tuning. default tuning methods include three hyperparameters tuning modes. The first mode involves keeping the hyperparameters unchanged. Most hyperparameters can be directly determined before model training and do not require tuning. For instance, the order of $\Theta_g$ is generally set to 2, and $k_1 = k_2 = 0.8, \gamma = 1$. The second mode is the conditional tuning method. It adjusts parameters based on the difference between the input data and global predictions, i.e., $\alpha, \beta$. When the mean difference between global predictions and input exceeds half of the current input data's semi-range, indicating a significant global prediction error that may affect local predictions, we set $\alpha = 0.8, \beta = 0.9$. Otherwise, we set $\alpha = 0.1, \beta = 0.9$. The third mode is the adaptive tuning method. It is $\gamma$ in $l_1$ regularization. Considering that $\gamma$ being too large or too small will not yield good predictive results, we define a feasible range for $\gamma$, such as $[1e - 7, 1e - 1]$. We employ binary search on the validation set to find a potentially optimal $\gamma$ value. Moreover, GLIP provides manual tuning methods so that you can also choose to adjust the parameters according to your needs.

## E  Pseudo-code of GLIP

---

**Algorithm 1:** Global Local Identification and Prediction (GLIP)

---

**Input:** Time-series $\mathbf{X}$, Iutput length $I$, Output length $O$, hyperparameter $\alpha, \beta, \gamma$, etc.

**Output:** Global predicction $\mathbf{X}^*_{gp}$, Local prediction $\tilde{\mathbf{X}}^*_{\text{pred}}$

1  /* **Global Basis and Global Prediction** */
2  Do DFT on the entire training set and get $\mathbf{A}, \mathbf{f}$, then get $\mathbf{T}^*$.
3  Create global basis $\Theta_g$ and compute equation 2, then get global predict $\mathbf{X}_{gp}$.
4  Create $\Theta^*_g$ with $\mathbf{X}_{gp}$ and compute equation 2 again, and get $\mathbf{X}^*_{gp}$ as output.
5  /* **Storage Basis** */
6  Create a set $\mathbf{T}^*_s$.
7  **for** *Each batch in training set* **do**
8  $\quad$ Conduct DFT on each batch and obtain $\mathbf{A_2}, \mathbf{f_2}$, then gain $\mathbf{T}^*_2$.
9  $\quad$ $\mathbf{T}^*_s \leftarrow \mathbf{T}^*_s \bigcup \mathbf{T}^*_2$.
10  Create storage basis $\Theta_s$ with $\mathbf{T}^*_s$.
11  /* **Validation Process** */
12  Validate the effectiveness of global predictions and the relationships between variables on the validation set and determine their suitability for further application in local predictions.
13  /* **Local Basis and Local Prediction** */
14  **if** *Global prediction is validated* **then**
15  $\quad$ Use global prediction as local prediction and get $\tilde{\mathbf{X}}_{\text{pred}}$.
16  **else**
17  $\quad$ **for** *Each batch in the test set* **do**
18  $\quad\quad$ Weight input data with equation 6.
19  $\quad\quad$ Conduct DFT on the weighted input data and obtain local basis $\Theta_l$.
20  $\quad\quad$ Create basis $\Theta_{\text{pred}}$ with equation 8 for local prediction and obtain $\tilde{\mathbf{X}}_{\text{pred}}$.

21  **if** *The relationships between variables are validated* **then**
22  $\quad$ Use the relationships between variables to form a basis for identification and once again, resulting in local prediction outputs $\tilde{\mathbf{X}}^*_{\text{pred}}$.
23  **else**
24  $\quad$ Output the local prediction results $\tilde{\mathbf{X}}_{\text{pred}}$.

---

