# OpenReview forum: "Interpretable Sparse System Identification: Beyond Recent Deep Learning Techniques on Time-Series Prediction"
_ICLR.cc/2024/Conference — ICLR 2024 poster_

### Official Review · Reviewer_fA8m · 2023-10-24

**Soundness:** 3 good
**Presentation:** 3 good
**Contribution:** 4 excellent
**Rating:** 8
**Confidence:** 2

**Summary:**

The work proposes Global-Local Identification and Prediction (GLIP) for time series forecasting. The proposed method moves away from deep learning (often transformer-based) methods, instead utilising lightweight approaches that can be run on a CPU (system identification and Fourier transformation). This has the further advantage of making training time (mostly) independent of the prediction horizon. In an evaluation over four datasets, GLIP shows improved performance for both MSE and MAE compared to seven other methods.

**Strengths:**

The proposed method is a strong combination of global prediction (forecasting the entire time series) with local information (rolling batches). As shown in Figure 2, the method is able to weight global/local information when most appropriate. Results compared to baselines in Table 1 are strong across all output lengths and datasets. This is especially impressive given the speed of the method ("seconds to minutes").

**Weaknesses:**

**Clarity**
C1. Inline citations are not in brackets, which sometimes makes the text hard to follow.
C2. Acronyms are often not defined.
C3. There are some typos throughout the work.

**Quality**
Q1. Percentage improvement is only given for MSE. Additional analysis of percentage improvement for MAE would be helpful (potentially in the appendix).
Q2. I have a small concern regarding data leakage from the test set. I believe the test batch information is only used for local prediction (constructing the storage basis), but in Figure 1 there is an arrow going from the test batch back into the global prediction (predicted length). Could the authors please verify that there is no data leakage that would affect the global prediction/overall results.

**Questions:**

1. How are values for $\alpha$ and $\beta$ selected? The work explains the what high/low values mean and when they should be chosen, but how were they determined in practice?
2. Section 4.1.3 gives a vague measure of "seconds to minutes" for running the experiments. Are the authors able to provide concrete training/inference times for GLIP, and compare this to the same for the baselines?

---

> ### Author Response · Authors · 2023-11-16
> **Response to Comments from Reviewer fA8m**
>
> Thanks for valuable comments and positive support. Below are the responses point by point:
> > **_Clarity_**
> * **C.1 Inline citations are not in brackets:** We have added brackets for each citations to improve the clarity.
>
> * **C.2 Acronyms are often not defined:** We have included the full names of relatively less familiar models such as TRMF (Temporal Regularized Matrix Factorization) and TLAE (Temporal Latent Auto-Encoder) in the manuscript when they first occured. For commonly used models like RNN, SVM, LSTM, GRU, etc., due to space constraints, we still keep using the abbreviations.
>
> * **C.3 Some typos throughout the work:** We have thoroughly checked the entire manuscript to reduce typos and improve presentation. For example, modifying the formula (3) in section 3.1.1, rectifying instances in Chapter 3 where "training set" was erroneously written as "test set," and resolving language issues in section 4.1.4. Moreover, corrections have been made for some capitalization and spelling errors.
>
>
> > **_Quality_**
> * **Q.1 Percentage improvement of MAE:** Thanks for suggestion. We have put them in Appendix D.3.
>
> * **Q.2 Concern regarding data leakage:** We apologize for the confusion caused by the incorrect use of arrows in Figure 1. The arrow associated with "Prediction Length" indicates that we input the length of the local prediction into the training set. Subsequently, we perform the DFT on the data in the training set with this specified length, forming a basis for storage. This involves only inputting the length value, for instance, predicting 96 steps would entail inputting the value 96, without including any values from the test set. Hence, there is no issue of data leakage. We have modified the arrows labeled "Prediction Length" to indicate directly from the dataloader since the data loader can provide the output length. We believe this reduces any further misunderstandings.
>
> > **_Questions:_**
> * **1. Hyperparameters:** In Appendix D.4, we have provided a detailed explanation of the hyperparameter selection process. The hyperparameters $\alpha$ and $\beta$ are conditionally tuned. When the mean difference between global predictions and input exceeds the half range of the input data, indicating a significant global prediction error that may affect local predictions, we set $\alpha=0.8, \beta=0.9$. Otherwise, we set $\alpha=0.1, \beta=0.9$. There are two benefits to this approach to adjust $\alpha$ and $\beta$. On the one hand, $\alpha$ and $\beta$ are not very sensitive to prediction results, so they do not require too much fine-tuning, since conditional tuning process makes it efficient. On the other hand, it prevents the disruption of local predictions by bad global prediction results.
>
> * **2. Concrete times for GLIP:** Specific computation time for GLIP and neural networks is as follows. For instance, on the ETTh dataset, the computation time is as follows. The unit after the values is second.
>
> | Prediction Length | GLIP | DLinear | Autoformer |
> |:---------------:|:--:|:-:|:-:|
> |      96      |  116s  |  189s  |  299s  |
> |      192     |  122s  |  216s  |  325s  |
> |      336     |  138s  |  239s  |  378s  |
> |      720     |  139s  |  244s  |  412s  |
>
> Here, GLIP operates on a CPU enviornment, whereas the other two neural network models were run with support of GPU, Nvidia RTX A6000. Even under such conditions, our processing speed remains superior to that of neural networks. This illustrates the advantage of GLIP from the prespective of running time.

---

> > ### Comment · Reviewer_fA8m · 2023-11-20
> > **Follow Up to Rebuttal for Reviewer fA8m**
> >
> > Thank you the response to my review. My questions have been answered and appropriate adjusts to the work have been made.
> >
> > I do have one minor follow up regarding the computation time. I understand the advantage of using GLIP in a CPU environment (removing the need for GPUs). However, could further improvements in computation time be achieved by adapting the method to run on a GPU? I'm curious if the authors think even faster computation of GLIP is possible (but this is not a necessity for the work).
> >
> > Thank you again for your response.

---

> > > ### Author Response · Authors · 2023-11-20
> > > **Response to the new question from Reviewer fA8m**
> > >
> > > Thank you very much for the response. Such a comparison on computation time mainly suggests that GLIP would support long-term prediction tasks of time series with a low-cost device environment. Indeed, regarding the issue of computation time, the advantage of GPUs lies in handling large-scale parallel computing tasks, such as deep learning under the backpropagation mechanism, where GPUs typically provide much higher computational performance and efficiency. However, not all types of computing tasks benefit from the parallel processing capabilities of GPUs, especially when dealing with sequentially executed tasks or tasks requiring a substantial amount of branching. As GLIP primarily involves sequential execution processes, we suggest that even with GPU training, it may not necessarily accelerate the GLIP model to some degree. Please let us know if you have any further questions.

---

> > > > ### Comment · Reviewer_fA8m · 2023-11-20
> > > >
> > > > Thank you for the quick response. I assumed that would be the case, but thought it worth clarifying.

---

> > > > > ### Author Response · Authors · 2023-11-22
> > > > > **Response to Reviewer fA8m**
> > > > >
> > > > > Thank you again for all these insightful comments and the kind feedback.

---

### Official Review · Reviewer_jLT6 · 2023-10-29

**Soundness:** 4 excellent
**Presentation:** 3 good
**Contribution:** 4 excellent
**Rating:** 8
**Confidence:** 4

**Summary:**

The paper proposes a time series forecasting approach based on sparse identification of global, storage, and local basis functions.
In contrast to the recent transformer-based advances, this approach runs on CPU and is computationally highly competitive to other models. It also outperforms other Transformer-based methods by a large margin.

Overall, it integrates knowledge about the challenges in time series forecasting (concretely, incorporating global and local aspects of time series datasets). This is further highlighted by additional design decisions like extracting local basis functions, through a prediction-data weighted input time series to mitigate outliers and data-variations.
An ablation study stresses the importance of each of the designed components and justifies each aspect of the complex model design (Fig 1).

In my opinion and after reading the method and appendix carefully, I see this as an excellent paper that demonstrates how to obtain state-of-the-art results in an orthogonal way to mainstream Transformer deep-learning approaches recently proposed.

While the overall presentation is complete, well justified, and technically sound, the language seems over complicated at places and could/should be revised. For instance, the sentence
> "The parameters Ξ, procured through the sparse identification process, exhibit the capacity to signify the intensity of latent cycles and the magnitude of inter-variable influences. By virtue of these parameters, a more profound comprehension of the model can be attained. "

could be rewritten in simpler (and more understandable) English by

> "The parameters Ξ indicate the cycles and magnitudes between variables, which can be used directly to interpret the model."

Similarly, pointing to individual panels in Figure 1 while describing the method would also greatly support the understanding of the method.
The current version describes the process in text and just references the figure as a whole.

**Strengths:**

* strong state-of-the-art performance across different forecasting benchmarks
* strong computational efficiency (without GPUs)
* extensive ablation study to justify each of the (many) components in the approach

**Weaknesses:**

* complex language makes the text more arduous to understand than necessary
* no details on hyperparameter tuning provided in the appendix.

**Questions:**

* what do the authors mean exactly with "reasonable" in their statement: code is available upon **reasonable** request? Will the source code be published on GitHub? If not, what is the motivation behind not providing the code open source?
* eq 3: how many higher-order polynomials ($X^2$, $X^3$, etc) were integrated in the global features?
* Can the authors please provide a detailed description of how the hyperparameters were obtained? The appendix/SI does not provide any information in this regard. Were they manually set or obtained by random search? What was the training/validation/evaluation split? How long did the hyperparameter tuning process take? How sensitive is the overall model to the choice of parameters? Were one set of hyperparameter set for every dataset or were some identical hyperparameters used across all benchmark datasets? This information is not in the current appendix!

---

> ### Author Response · Authors · 2023-11-16
> **Response to Comments from Reviewer jLT6**
>
> Thanks for valuable comments and positive support. Below are the responses point by point:
> > **_Weakness_**
>
> * **W.1 Language issues:** We have carefully checked the language issues in subsection 4.1.4. Additionally, we have rephrased the cotents in Chapter 3. To enhance clarity in the new version, we have added multiple citations referring to individual panels in Figure 1 in subsections 3.1.1, 3.1.2 and 3.3.2. Moreover, we have included pseudocode for the entire algorithm in Appendix. E to facilitate a better understanding for the readers.
>
> * **W.2 Details on hyperparameter tuning:** We have relocated the hyperparameter tuning process to Appendix D. 4. For specific questions regarding certain hyperparameters, we have provided additional answers to Q.3 below.
>
> > **_Questions:_**
>
>
> * **Q.1 Code:** Open-source code has been provided in the supplementary material. It is convenient for users to reproduce the experiments. Thus, we have deleted the sentence, "code is available upon reasonable request" in the new version.
>
> * **Q.2 The order of polynomials in global features:** In our manuscript, we just provided up to second-order polynomial relationships, as explained in detail in Appendix D.4. This is due to the fact that excessively high-order polynomial relationships may lead to the issue of dimensionality explosion and overfitting.
>
> * **Q.3 Detailed description of hyperparameters:** In Appendix D.4, we have supplemented explanations for the selection of most hyperparameters. GLIP has three parameter tuning modes based on different parameter attributes. Below are the point-by-point responses to the questions:
> 1.  The first mode involves keeping the hyperparameters unchanged. Most hyperparameters can be directly determined before model training and do not require further tuning. For instance, the order of $\mathbf{\Theta}_g$ is generally set to be 2, and $k_1=k_2=0.8, \gamma =1$.
> 2. The second mode is the conditional tuning method. It adjusts hyperparameters based on the difference between the input data and global predictions, i.e., $\alpha, \beta$. When the mean difference between global predictions and input exceeds the half range of the input data, indicating a significant global prediction error that may affect local predictions, we set $\alpha=0.8, \beta=0.9$. Otherwise, we set $\alpha=0.1, \beta=0.9$. The setting for this parameter varies across different datasets. While, in fact, even if one adjusts the above two modes of hyperparameters, the impact on predictions is not sensitive (sensitivity experiments will be discussed in point 6 here).
> 3. The third mode is the adaptive tuning method which is also the main parameter to adjust, i.e., $\gamma$ in $l_1$ regularization. Considering that $\gamma$ being too large or small will not yield good predictive results, we define a feasible range for $\gamma$, such as $[1e-7, 1e-1]$. We employ binary search on the validation set to find a potentially optimal $\gamma$ value. The setting for this parameter varies across different datasets.
> 4. The ratios for the training set, validation set, and test set were mentioned at the beginning of Chapter 3, and they are seperated in  proportion of 7:1:2. The original content is as follows: "Analogous to deep learning methodologies, the time series will be partitioned into distinct sets, which are training set, validation set, and testing set, with a partitioning ratio of 7:1:2."
> 5. Due to the small size of the validation set and fast speed of GLIP, hyperparameter determination usually takes less than one minute.
> 6. The given hyperparameters are not sensitive to variations in the data. Regarding the Exchange data, we modified the values of $\alpha, \beta,\gamma$, and the results of the sensitivity experiment are as follows (The values in the table represents MAE/MSE.):
>
> | hyperparameters | $\alpha$ | $\beta$ | $\gamma$ |
> |:----:|:-:|:-:|:-:|
> |      origin      | 0.443/0.516  |  0.443/0.516 | 0.443/0.516  |
> |      origin±0.05      | 0.441/0.515  | 0.433/0.512  |  0.440/0.514  |
> |      origin±0.1      |  0.415/0.501  | 0.425/0.508  |  0.441/0.515  |
> |       origin±0.2      |  0.453/0.523  |  0.423/0.507  |  0.442/0.515  |
>
> From the above table, it can be observed that even with certain variations in these hyperparameters, the experimental results only show slight changes. The difference is approximately around 5%, and it still maintains state-of-the-art prediction performance. Therefore, GLIP is not sensitive to hyperparameters.

---

> ### Comment · Reviewer_jLT6 · 2023-11-21
> **Follow Up to Rebuttal; No further questions**
>
> On the source code, thank you for proving the code in the supplementary material.
>
> The response to the hyperparameter tuning is also reasonable.
>
> Thank you for the reply to my raised questions. I have no further questions here.

---

> > ### Author Response · Authors · 2023-11-22
> > **Response to Reviewer jLT6**
> >
> > Thanks very much for the insightful and constructive comments and the positive recommendation.

---

### Official Review · Reviewer_sU4J · 2023-11-08

**Soundness:** 2 fair
**Presentation:** 1 poor
**Contribution:** 3 good
**Rating:** 6
**Confidence:** 3

**Summary:**

The paper presents  Global-Local Identification and Prediction (GLIP), a new approach for long-term time-series prediction based on sparse system identification. The proposed approach utilizes discrete Fourier transformation in global identification and local prediction and the experiments show significant improvement over the baselines, in particular in long-horizon prediction.

**Strengths:**

Strengths:
- Novel approach for long-term time-series prediction
- Very promising results showing significant improvement over the state of the art
- The method performs particularly well for longer horizons, with the higher improvement compared to baselines on the longer horizons

**Weaknesses:**

I found the description of the method very hard to follow with many details either missing or unclear.
- Section 3.1.1:
	* while the section describes a Discrete Fourier Transformation (DFT) on X, it is not clear where in this section the resulting A,f are used.
	* it is not clear what T/T* are or how are they computed? Is \Theta_g includes exactly three items or multiple sin/cos waves? (this question also applied to Section 3.1.2)
	* "from the univariate predictions with the test set" - should probably be the training set?
	* X^*_g is not really used in Eq. (3), different from the text. Also, not clear what order of expressions were considered.
- Section 3.2: it is not clear to me if this is part of the model/training or a description of how to validate the model's performance (e.g., the paper mentioned it is used to "evaluate" or "conduct an assessment")?
- Overall, I think the paper would significantly benefit from a pseudo-code describing training and inference procedures.

One significant weakness is related to interpretability. The paper positions the proposed approach as an interpretable approach (including the title of the paper and in the motivation described in the abstract). However, the paper presents no results on interpretability in the paper and there is only a short description that says that the parameters are "amenable to interpretation". To make strong claims regarding the interpretability of the proposed approach, some results are needed (i.e., what are these parameters, are they weights associated with interpretable features, how many are there, etc). Some example(s) for the interpretability of results would be very useful as well.

Experiments:
- Datasets: the number of datasets used in the paper seems significantly smaller compared to comparable works. For example, in [2] and [3], other datasets such as weather, electricity, and traffic were included.
- Global prediction is presented as an advantage of the proposed approach, but without any baseline, it is very hard to judge whether the results we observe are good (e.g., in Exchange and ILI where the predicted patterns deviate from the ground truth). It would be useful to include some time-series forecasting baseline so that we can get a sense of how well the proposed approach is doing by comparison.
- Reproducibility: missing details as described above (e.g., the construction of \theta_g). In addition, the chosen values for the hyper-parameters (e.g., \alpha, \beta) are not described (as well as the procedure of tuning them).

Literature: the work should probably mention other models that have focused on basis expansion, such as N-BEATS [1].

Minor issues:
- Section 3: " in the testing set for global prediction" -> should be training set?
- page 6: "prediction. we aim" -> "prediction. We aim"


[1] Oreshkin, B. N., Carpov, D., Chapados, N., & Bengio, Y. (2019, September). N-BEATS: Neural basis expansion analysis for interpretable time series forecasting. In International Conference on Learning Representations.
[2] Wu, H., Xu, J., Wang, J., & Long, M. (2021). Autoformer: Decomposition transformers with auto-correlation for long-term series forecasting. Advances in Neural Information Processing Systems, 34, 22419-22430.
[3] Zeng, A., Chen, M., Zhang, L., & Xu, Q. (2023, June). Are transformers effective for time series forecasting?. In Proceedings of the AAAI conference on artificial intelligence (Vol. 37, No. 9, pp. 11121-11128).

**Questions:**

I would appreciate the authors' response to the weaknesses listed above.

---

> ### Author Response · Authors · 2023-11-16
> **Response to Comments from Reviewer sU4J (part I)**
>
> Thanks for your meticulous reading of our article and providing us with valuable suggestions. We have carefully considered these comments and suggestions. We have categorized the weaknesses and questions you pointed out into the following three aspects: details regarding the article, descriptions of interpretability, and the presentation of experiments. Below is the response point by point to the questions.
>
> > **_W1/Q1 Details regarding the article_**
> * **W/Q 1.1 The utilization of $\mathbf{A}$ and $\mathbf{f}$ in *DFT.***
>
> **A**: Through the application of ***DFT***, we can transform time series into the frequency domain, where $\mathbf{A}$ represents the amplitude and $\mathbf{f}$ signifies the frequency. $\mathbf{A}$ and $\mathbf{f}$ are vectors with the equal length, maintaining a one-to-one correspondence. The larger the amplitude, the greater the impact of the corresponding frequency on the time series. Our objective is to select, based on the amplitude magnitude, the frequencies most likely to construct the main components of the time series. Subsequently, these selected frequencies are transformed into basis functions for consideration, which will be further screened through sparse identification. Therefore, our candidate periodical terms $\mathbf{T^*}=[T_1, T_2,...,T_{m_1}]$ are determined by the selection process involving $\mathbf{A}$ and $\mathbf{f}$.  We have incorporated this selection process into Appendix. C and referenced it in the main text after the statement "For further global prediction, frequencies with high amplitude are selected" in the new version.
>
> * **W/Q 1.2 The computation of $\mathbf{T}, \mathbf{T^{*}}, \mathbf{\Theta_g}$.**
>
> **A**: After a series of potential frequencies $\mathbf{f}$ influencing the time series were selected based on the higher amplitudes $\mathbf{A}$, a series of potential periods $T_i$ were subsequently derived by employing the frequency-period conversion formula $f=1/T$. This forms a row vector $\mathbf{T^*}=[T_1, T_2,...,T_{m_1}]$, where each element records a potential period most likely to influence the given time series. Similarly, $\mathbf{C}_1$ is also a row vector, as mentioned in the Notation section regarding the computation of operations involving scalars and vectors. The construction of matrix $\mathbf{\Theta_g}$ is not limited to three basic items/functions. Taking $\sin(\mathbf{C}_1\mathbf{t})$ as an example, where $\mathbf{C}_1$ is an $m_1 \times 1$ row vector and $\mathbf{t}$ is a $1 \times n$ column vector, they form an $m_1 \times n$ matrix. Thus, $\sin(\mathbf{C}_1\mathbf{t})$ is also an $m_1 \times n$ matrix. The construction of $\sin(\mathbf{C}_2\mathbf{t})$ is similar, and the final $\mathbf{1}$ represents a column vector of all ones. Therefore, there should be a total $2m_1 +1$ basic functions. This explanation is equally applicable to sections 3.2 and 3.3. To enhance clarity, when $\mathbf{T^*}$ and $\mathbf{t}$ are first introduced in Section 3.1, we have added row and column vector labels for them in the new version.
>
> * **W/Q 1.3 Typographical error:**
>
> **A**: In the old version, indeed, "from the univariate predictions with the test set," should be "from the univariate predictions with the training set". We have corrected it in the new version.
>
> * **W/Q 1.4 Problem in $\mathbf{\Theta}_g^{*}$:**
>
> **A**: The functions formed by $\mathbf{X}$ consisting the expression of $\mathbf{\Theta}_g^*$ in Eq. (3) should be $\mathbf{X}^*$. We have corrected it in the new version. The order of expressions is 2. This will be explained in Appendix (D.4).
>
> * **W/Q 1.5 Problem in Section 3.2:**
>
> **A**: In Section 3.2, our validation set is used to evaluate the effectiveness of the training set. The results from the validation set here will impact the subsequent use of local rolling predictions. We have provided pseudocode in Appendix. E, and its role can be inferred from line 14 and line 21 of the pseudocode. Our approach follows the neural network paradigm, dividing the data into training, validation, and test sets. Here, the role of the validation set is similar to that in neural network methods.  To enhance clarity in our expression, we have rephrased the description in subsection 3.2.1 and rewritten subsection  3.2.2. We hope that, such a revision could help improve the understanding of the role of the validation set in GLIP.
>
> * **W/Q 1.6 Pseudo-code**
>
> **A**: Thanks for the suggestion. We have given a pseudo-code in the Appendix. E to improve the presentation, and we referenced it after introducing the method in subsection 3.3.3.

---

> > ### Author Response · Authors · 2023-11-16
> > **Response to Comments from Reviewer sU4J (part II)**
> >
> > > **_W2/Q2 Descriptions of interpretability_**
> >
> > **A**: We suggest interpretability pervades GLIP and it can be explained from three aspects. Firstly, GLIP employs DFT to explicitly identify potential periods within the time series for both global and local identification, incorporating them into the basis functions. This can be utilized to interpret the main components extracted from the frequency domain of the time series. Secondly, in comparison with neural networks, GLIP has an explicit mathematical function expression for time series prediction tasks. The function expression provides insights into the predicted trends with an explicit white-box form. It is worth noting that in the article of N-BERTS model mentioned in the review, the "Interpretability" section in N-BERTS also discusses the explicit representation of trend model and seasonality model. Thirdly, the hyperparameters, namely $k_1, k_2, \alpha, \beta,\lambda$, and hyperparameters $\Xi$ have practical significance and can be used directly to interpret the model. Comparatively, hyperparameters such as the number of hidden layers and neurons in neural networks or the number of multi-heads in Transformer may not have straightforward interpretability. Overall, we believe that GLIP, at least compared to most neural network based prediction methods, possesses interpretability to a large degree. We have further elaborated on this point in subsection 4.1.4 in the new version.
> >
> > > **_W3/Q3 The presentation of experiments_**
> > * **W/Q 3.1 Dataset numbers**
> >
> > **A**: GLIP has one advantage that the computational capability is supported by CPU so as to be more suitable for datasets with lower variable dimensions, since we can use sufficient candidate functions to model the inter-coupling relationships between these variables. For datasets with a high dimension of variables, constructing finite basis functions becomes a big challenge in accurately capturing the underlying mechanistic features of the data. Additionally, as the dimension of variables increases, the composition scale of basis functions grows exponentially, leading to the curse of dimensionality in optimization computations. GPU does not support the algorithm for optimization process of system identification in datasets with a very high dimension of variables. That is, system identification algorithms do not rely on computational power requirements to some extent, which cannot benefit from the powerful computation support from GPU. Therefore, deep learning algorithms, supported by GPU, may demonstrate advantages for datasets with a very high dimension of variables. E.g., in the datasets with less than twenty dimensions of variables presented in this study, GLIP exhibits clear advantages. However, as the number of variables increases, modeling such time series with hundreds of thousands of basis functions (multipliers) becomes a huge challenge for accurately reconstructing and predicting the systems. We have mentioned this in the Conclusion and Future work section in the old version. However, this issue is not insurmountable. One solution is that we have tested GLIP as a pre-trained module, integrated into the architecture of neural network. This integration widely improves reconstruction and prediction effects. Yet, this integrated approach sacrifices the interpretability and low computational requirements inherent in standalone system identification algorithms. Therefore, this study does not delve into the detailed exploration of this integrated architecture. Another solution is that we randomly selected ten inter-coupling variables for the target variable in the multi-variable dataset with more than e.g., twenty variables. Such a slight implement change of GLIP also indicates advantages compared with Dlinear and Autoformer, see the following Table. Here, the bold numbers represent the best prediction results, and the italic numbers mean the second-best results. Additionally, with a better selction mechanism of inter-coupling variables, the performence of GLIP for large-scale multi-variable datasets can be further improved.

---

> > > ### Author Response · Authors · 2023-11-16
> > > **Response to Comments from Reviewer sU4J (part III)**
> > >
> > > Weather:
> > > | Prediction Length | GLIP(MAE/MSE)   | DLinear(MAE/MSE) | Autoformer(MAE/MSE) |
> > > |:--------------:|:------:|:-------:|:----------:|
> > > |       96       |    **0.159**/**0.226**    |    0.176/0.237     |    0.266/0.336        |
> > > |       192      |    **0.164**/**0.271**    |    0.220/0.282     |     0.307/0.367        |
> > > |       336      |    **0.248**/0.348    |    0.265/**0.319**     |     0.359/0.395       |
> > > |       720      |    **0.316**/0.404    |    0.323/**0.362**     |     0.419/0.428       |
> > >
> > > Electricity:
> > > | Prediction Length | GLIP(MAE/MSE)   | DLinear(MAE/MSE) | Autoformer(MAE/MSE) |
> > > |:--------------:|:------:|:-------:|:----------:|
> > > |       96       |    **0.140**/*0.243*    |    **0.140**/**0.237**     |      0.201/0.317      |
> > > |       192      |    *0.162/0.267*    |    **0.153**/**0.249**     |      0.222/0.334      |
> > > |       336      |    **0.166**/*0.309*    |    *0.169*/**0.267**     |      0.231/0.338      |
> > > |       720      |    **0.180**/**0.294**    |    *0.203/0.301*     |      0.254/0.361      |
> > >
> > > Traffic:
> > > | Prediction Length | GLIP(MAE/MSE)   | DLinear(MAE/MSE) | Autoformer(MAE/MSE) |
> > > |:--------------:|:------:|:-------:|:----------:|
> > > |       96       |    **0.402**/*0.294*    |    *0.410*/**0.282**     |    0.613/0.388        |
> > > |       192      |     **0.421**/*0.289*   |    *0.423*/**0.287**     |    0.616/0.382        |
> > > |       336      |    **0.428/0.295**    |    *0.436/0.296*     |    0.622/0.377        |
> > > |       720      |    **0.450**/*0.325*    |    *0.466*/**0.315**     |   0.660/0.408         |
> > >
> > > * **W/Q 3.2 Global Prediction**
> > >
> > > **A**: The reason for the absence of a Baseline in global predictions is that, up to now, we have not encountered any neural network models capable of predicting time series beyond 1000 steps. As mentioned in Appendix. A, neural networks require a sufficient number of samples for training, and achieving extended predictions poses a challenge in terms of both training duration and sample quantity. Consequently, it is impractical to compare with neural networks on global prediction. For classical methods such as ARIMA, when making long-term predictions, significant error accumulation occurs, leading to substantial errors, rendering the comparison meaningless.
> > >
> > > * **W/Q 3.3 Reproducibility**
> > >
> > > **A**: Regarding the hyperparameter settings, we have provided detailed descriptions in Appendix D.4, including the hyperparameters of your concern, namely $\mathbf{\Theta}_g, \alpha, \beta$.
> > >
> > > > **_Additional Q/W: Literature & Minor issues_**
> > > * **Literature**
> > >
> > > **A**: Thanks for the suggestion. We have cited and accordingly added an explanation for interpretability in the fourth paragraph of the new Introduction section.
> > >
> > > * **Minor issues**
> > >
> > > **A**: Revised with many thanks.

---

> > > > ### Comment · Reviewer_sU4J · 2023-11-23
> > > >
> > > > Thank you for your comments. Some of my concerns have been addressed. I do think all the additional information provided in the response should be reflected in the paper. However, I am still not convinced that the proposed approach provides significant benefits in terms of interpretability.
> > > >
> > > > I have decided to increase my score by one point.

---

> > > > > ### Author Response · Authors · 2023-11-23
> > > > > **Response to Comments from Reviewer sU4J**
> > > > >
> > > > > Many thanks for the kind response and the acknowledgement of our responses. Interpretability is a fundamental issue at the forefront of artificial intelligence. We would like to provide an alternative solution with explicit mathematical formula here to the long-term prediction issue, compared with typical deep learning approaches. In our future work, we are eager to further explore the interpretability of the GLIP and the related model framework to make it more intuitive and convincing accordingly. At the same time, we will continue to enhance the accuracy and efficiency of the model, striving for further breakthroughs in lightweight models.

---

### Meta-Review · Area_Chair_ScUu · 2023-12-08

**Metareview:**

The reviewers reached an agreement for acceptance, and I am pleased to recommend this paper based on their expertise. It is essential that the authors address the reviewers' concerns in the final version, giving particular attention to the concerns raised by Reviewer sU4J.

**Justification For Why Not Higher Score:**

The interpretability of this method is not as positioned as the paper claims and needs calibration.

**Justification For Why Not Lower Score:**

This paper is a good contribution due to the novel idea and sota results.

---

### Decision · Program_Chairs · 2024-01-16

Accept (poster)